# Mycotoxin Decontamination Efficacy of Atmospheric Pressure Air Plasma

**DOI:** 10.3390/toxins11040219

**Published:** 2019-04-12

**Authors:** Nataša Hojnik, Martina Modic, Gabrijela Tavčar-Kalcher, Janja Babič, James L. Walsh, Uroš Cvelbar

**Affiliations:** 1Laboratory for Gaseous Electronics F6, Jozef Stefan Institute, Jamova 39, 1000 Ljubljana, Slovenia; martina.modic@ijs.si (M.M.); uros.cvelbar@ijs.si (U.C.); 2Jožef Stefan International Postgraduate School, Jamova 39, 1000 Ljubljana, Slovenia; 3Institute of Food Safety, Feed and Environment, Veterinary Faculty, University of Ljubljana, Gerbičeva 60, 1000 Ljubljana, Slovenia; gabrijela.tavcar-kalcher@vf.uni-lj.si (G.T.-K.); Janja.Babic@vf.uni-lj.si (J.B.); 4Department of Electrical, Engineering and Electronics, University of Liverpool, Liverpool L69 3GJ, UK; J.L.Walsh@liverpool.ac.uk

**Keywords:** mycotoxins, cold atmospheric pressure plasma, decontamination

## Abstract

Mycotoxins, the toxic secondary metabolites of mould species, are a growing global concern, rendering almost 25% of all food produced unfit for human or animal consumption, thus placing immense pressure on the food supply chain. Cold Atmospheric pressure Plasma (CAP) represents a promising, low-cost, and environmentally friendly means to degrade mycotoxins with negligible effect on the quality of food products. Despite this promise, the study of CAP-mediated mycotoxin degradation has been limited to a small subset of the vast number of mycotoxins that plague the food supply chain. This study explores the degradation of aflatoxins, trichothecenes, fumonisins, and zearalenone using CAP generated in ambient air. CAP treatment was found to reduce aflatoxins by 93%, trichothecenes by 90%, fumonisins by 93%, and zearalenone by 100% after 8 minutes exposure. To demonstrate the potential of CAP-mediated mycotoxin degradation against more conventional methods, its efficiency was compared against ultraviolet C (UVC) light irradiation. In all cases, CAP was found to be considerably more efficient than UVC, with aflatoxin G_1_ and zearalenone being completely degraded, levels that could not be achieved using UVC irradiation.

## 1. Introduction

Mycotoxins are toxic compounds produced during the secondary metabolism of filamentous fungi and are known to regularly spoil food and feed products. Presently, more than 400 structurally different mycotoxins have been identified, including aflatoxins (AFs), ochratoxins, fumonisins (FBs), patulin, zearalenone (ZEN), ergot alkaloids, and trichothecenes, including deoxynivalenol (DON), HT-2 and T-2 toxin [1]. The recent and rapid increase in mycotoxin food contamination represents one of the main concerns in the field of agriculture and food production [2]. 

Most mycotoxins are chemically and thermally stable during food processing, meaning such contamination is extremely difficult to remove. They can harm human health through a wide range of toxic effects, including carcinogenicity, teratogenicity, hepatotoxicity, mutagenicity, neurotoxicity effects, and the disruption of both immune and endocrine systems [3]. In essence, the food supply chain urgently needs a mycotoxin decontamination method that is low-cost, highly effective, and can be applied to contaminated food to minimise waste and enhance safety for the consumer. Current management strategies for controlling mycotoxin occurrence are mostly based on chemical, biological, and physical approaches [4]. Amid these, ultraviolet (UV) irradiation is one of the most frequently used non-thermal food processing methods [5]. Several studies have identified that UV irradiation is a potential technology for the decontamination of mycotoxins such as AFs, trichothecenes, and ZEN [6,7,8,9]. Nevertheless, degradation efficiencies can vary widely due to differences in irradiation conditions, meaning the exposure times that are necessary to achieve a satisfactory level of decontamination are often too long to be relevant for large-scale food production.

Recently, exciting preliminary results have shown that Cold Atmospheric pressure Plasma (CAP) has considerable potential to reduce food contamination and improve food safety [10,11,12]. Plasma is the fourth state of matter or quasi-neutral ionized gas, primarily composed of photons, ions, free electrons, as well as atoms in their ground or exited states with a net neutral charge. The strong electric fields used in generating cold plasma accelerate electrons to energies capable of ionizing, electronically exciting, dissociating, and heating the constituents of the background gas. In air, this results in over 50 distinct chemical species, including the ionic species of H^+^, H^−^, O^+^, O^−^, H_3_O^+^, OH^−^, and N_2_^+^; the excited and ground states of N_2_, O_,_ NO, and OH; and longer-lived species such as O_3_, NO_2_, and N_2_O; collectively, these are referred to as reactive oxygen and nitrogen species (RONS) [13]. The concept of treating food with non-thermal plasma for the purposes of microbial inactivation has been under consideration for well over a decade [14,15], and it has been shown that the RONS produced within plasma are highly effective antimicrobial agents that are also capable of degrading a wide variety of toxic compounds, including mycotoxins [16,17,18].

In this study, the efficiency of a scalable and low-cost CAP-based approach was compared with ultraviolet C (UVC) irradiation for the degradation of a diverse array of mycotoxins. Aflatoxins B_1_ (AFB_1_), B_2_ (AFB_2_), G_1_ (AFG_1_), G_2_ (AFG_2_), DON, 3-Acetyl DON (3-AcDON), 15-Acetyl DON (15-AcDON), diacetoxyscirpenol (DAS), HT-2, and T-2 were considered, as well as fumonisins B_1_ (FB_1_), B_2_ (FB_2_), and ZEN. The CAP system was based on a surface barrier discharge (SBD) configuration, using ambient air as the working gas and two different powers, leading to two different CAP chemistries [19,20]. To determine decontamination efficacy, the remaining mycotoxin concentration following CAP or UVC exposure was assessed using liquid chromatography coupled with tandem mass spectrometry (LC-MS/MS).

## 2. Results and Discussion

The presented study exposes the high decontamination potential of CAP against mycotoxins, which was especially evident when compared to the commonly used UVC irradiation. The efficiency of CAP mycotoxin removal was evaluated according to the current European Union (EU) legislation dictating the maximum permissible levels of mycotoxin residual on food products. Strict monitoring procedures are currently mandated across the EU for the mycotoxins AFB_1_, DON, OTA, and ZEN as well as the total combined amount of AFs, HT-2, and T-2 and FBs [21,22,23]. The maximum levels allowed in selected food products are listed in Table 1.

It is widely reported that AF are the most toxic mycotoxins, with a chemical structure that categorises them as members of the difuranocoumarins. Critically, AFB_1_ has been identified as one of the leading causes for liver cancer worldwide as a result of toxic effects such as mutagenicity and cytotoxicity [24]. For the experiments detailed here, AFs were deposited on clean glass coverslips with the following concentrations: 18 µg/kg of AFB_1_ and AFG_1_, and 4 µg/kg of AFB_2_ and AFG_2_. As shown in Figure 1, following CAP exposure, the concentrations of AFB_1_ were reduced to values lower than 0.006 µg/kg after only 30 s of exposure (Figure 1a). The most efficient CAP condition was found to be the low power, reducing AFB_1_ to 0.003 µg/kg (>99.99% decontamination). In both cases, the AFB_1_ concentration was reduced well below the maximum level allowed according to EU legislation (2 µg/kg). The performance of CAP decontamination was even higher in the case of AFG_1_ (Figure 1c), where complete decontamination was noted after 60 s of treatment with the high-power operated CAP. 

In contrast, Figure 1b,d reveals that AFB_2_ and AFG_2_ were more persistent against both decontamination approaches used in the study. Nevertheless, high-power CAP conditions led to the highest level of degradation, reducing AFB_2_ by 70% and AFG_2_ by 74%. Despite the incomplete degradation, the sum of AFs exposed to CAP for 480 s did not exceed the maximum level allowed in cereals, i.e., 4 µg/kg (Figure 2). UVC irradiation, on the other hand, initially induced a rapid decrease in AFs concentration over the first few seconds of treatment, after which no further degradation was observed, yielding a total reduction of less than 50%.

In contrast to AFs, trichothecenes are known to be particularly resistant to many decontamination approaches. They represent a large group of compounds with approximately 170 different types of metabolites with a common a tetracyclic-12,13-epoxy skeleton. Their toxicity is based on their ability to interfere with the synthesis of proteins, resulting in acute poisoning symptoms such as diarrhoea, vomiting, and nausea [25,26,27]. Notably, Figure 3 indicates that a significantly higher degradation efficiency was achieved following CAP treatments compared to UV irradiation. In the case of DON, the greatest reduction was achieved after 480 s of CAP treatment under high power conditions, with just 0.11 mg/kg of DON remaining from a starting concentration of 2.7 mg/kg (Figure 3a), which falls below the maximum allowable level for cereals (0.75 mg/kg). Low power plasma conditions did not reach a comparable decontamination efficiency; nevertheless, the degradation still exceeded that found with UVC irradiation. Furthermore, comparable decontamination trends were also observed in the case of 3-AcDON, 15-AcDON, HT-2, and T-2 (Figure 3b,c,e,f). Their initial concentration was 2.5 mg/kg with the exception of HT-2, which had the higher concentration of 3.5 mg/kg. A significant amount, 0.5 mg/kg, of both 15-AcDON and T-2 was detected following even the longest exposures CAP, indicating that they are some of the most persistent trichothecenes to CAP-based approaches. Furthermore, none of the approaches examined in this study were able to reduce the sum of HT-2 and T-2 under the maximum levels allowed in food products such as processed oats intended for direct human consumption (Figure 4). Nonetheless, CAP treatments reduced the amount of HT-2 and T-2 under the 2 mg/kg, which is the maximum permissible level allowed in oat milling products used for feed [23]. Among trichothecenes, DAS was the most sensitive to CAP (Figure 3d), reaching a 97% reduction after a 480 s treatment under high power CAP conditions. 

FBs are polyketides with characteristic 2-propane-1,2,3-tricarboxylic acids esterified to an aminopolyol chain. Through their strong resemblance to sphinganine, they can disrupt cell functions, which involve sphingolipids, including cell proliferation, cell differentiation, and apoptosis. Belonging to a class of non-genotoxic carcinogens, they contribute to both liver and kidney cancers [28]. In this study, the starting concentrations of FBs were 2.7 mg/kg for FB_1_ and 2.9 mg/kg for FB_2_. In both CAP cases, a degradation in excess of 90% was observed (Figure 5), significantly higher than that achievable with UVC exposure. 

Figure 6 shows the sum of both FBs. Both CAP conditions reduced the summed concentration below the maximum permissible level for maize products, 1 mg/kg [22]. 

Finally, ZEN was considered: it has a 6-(10-Hydroxy-6-oxo-trans-1-undecenyl)-β-resorcylic acid lactone structure that has shares similarities with the human sex hormone 17β-estradiol. Consequently, it can bind to oestrogen receptors, leading to endocrine disruption and reproduction issues [29]. In this study, an initial ZEN concentration of 2.5 mg/kg was used. Complete removal was observed following exposure to CAP after 60 s of treatment under high power conditions (Figure 7). Notably, only 45% degradation was observed after 480 s of UVC irradiation. Moreover, UVC irradiation was found to induce the formation of an additional chromatogram peak near the primary peak, likely resulting from photoisomeration (Appendix A). It is well known that ZEN exists in two stereoisomeric forms; in the presence of UV photons, the naturally occurring ZEN trans form can be converted to the more stable cis-ZEN [30]. In contrast, no cis-ZEN was observed after the treatment with CAP. 

A limited number of previous CAP studies have considered the decontamination of different kinds of mycotoxins. Given the highest toxicity among mycotoxins, majority of studies have explored the degradation of AFs [16,17,31]. A dielectric barrier discharge (DBD) system (AcXys Technologies, St. Martin Le Vinoux, France) was used for AFs removal with an N_2_ as carrier gas, resulting in 100% decontamination of AFB_1_ after 4 min of treatment [17]. As in our case, AFB_1_ and AFG_1_ were more sensitive to CAP compared to AFB_2_ and AFG_2_. Moreover, studies using UV irradiation as an AFs decontamination approach showed similarities with the findings reported in this study, demonstrating that significant levels of decontamination can only be reached with long exposures (>30 min) [6,8,32]. Apart from AFs, a study by ten Bosch et al. explored the efficiency of air-based DBD plasma in decontamination of various mycotoxins, including DON, T-2, FB_1_, and ZEN [18]. Compared to our results, the degradation rates varied with the type and structure of mycotoxin. 

To understand the decontamination based on CAP, the physical-chemical properties of CAP must be considered. CAP-induced chemical degradation is mostly initiated through the generation of the variety of RONS. Short-lived species such as atomic oxygen (O), hydroxyl radical (OH), atomic nitrogen (N), and superoxide ion (O_2_^−1^) are produced next to the plasma region. Following this, they are distributed away from the electrode, where they react with each other, forming long-lived species such as ozone (O_3_), hydrogen peroxide (H_2_O_2_), nitric oxide (NO), and other nitrogen oxides (NO_x_) [19]. Experiments done in this study were performed with the 5 mm distance between the mycotoxin sample and outer electrode, meaning that long-lived species contributed the most to the CAP degradation of mycotoxins. 

Low power conditions induce the formation of reactive oxygen species (ROS)-dominant plasma effluent chemistry with O_3_ as the primary long-lived element. On contrary, a significant rise of the reactive nitrogen species (RNS) is observed under high power conditions, including N_2_O and NO_2_, eventually resulting in O_3_ decomposition. Alternatively, plasma effluent can be also defined as chemistries containing low and high dose of RONS, corresponding to the low and high power conditions [19,20,33]. The interaction between aqueous interface of wet mycotoxin sample and long-lived RONS induces the formation of a special liquid chemistry, which reintroduces very reactive, indirectly produced short-lived species, including OH and peroxynitrite (ONOO^−^) [34,35]. 

CAP RONS chemical degradation mostly involves radical reactions. RONS have a tendency to react with the least stable sites of the molecules such as double bonds between two carbon atoms, especially with those that are geometrically more exposed. RONS double bond attack is carried out through different mechanisms, starting with hydrogen abstraction, epoxidation, and oxidation of the carbon atoms with bond break as another possible result [36,37]. Such double bonds are included in the chemical structures of AFB_1_, AFG_1_, and ZEN, which were the most sensitive to the CAP treatments. In contrast, the rest of the mycotoxins possess more stable structures and are therefore more resistant to the CAP RONS. In addition, samples with trichothecenes contained a significantly higher number of mycotoxin molecules compared to other mycotoxins, meaning that a larger number of RONS is necessary for their complete removal. Consequently, longer exposure times are required for higher decontamination rates.

## 3. Conclusions

Mycotoxin members AFs, trichothecenes, FBs, and ZEN were successfully decontaminated after exposure to the CAP generated RONS. CAP based on the highly scalable air SBD system is known to produce high concentrations of RONS already in short operating times, which can react with a large number of mycotoxin molecules at once, leading to considerably higher decontamination efficiency compared to UVC irradiation, which is one of the most frequently used food processing approaches. Regarding their short lifetime, RONS decompose shortly after their formation, leaving no residuals. 

## 4. Materials and Methods

### 4.1. Preparation of Mycotoxins and CAP Treatments

For the treatments, stock solutions of mycotoxins were prepared from their standards purchased from Romer (Union, MO, USA) in the mixture of acetonitrile and deionized water (70% acetonitrile/30% water (*v*/*v*)) in order to obtain better decontamination performance. The concentrations of mycotoxin stock solutions are listed in the Table 2. A volume of 50 µL of each mycotoxin solution was deposited on a glass coverslip and left for 5 minutes to dry. Immediately after, the glass coverslips with a layer of mycotoxin were exposed to CAP.

The CAP treatments were performed with highly scalable air SBD system similar to that reported by Modic et al. [14]. The plasma was generated from specially fabricated electrode unit made of 1 mm thick quartz disc placed between two electrodes and plastic housing. The inner copper electrode was connected with high voltage source, covered with epoxy resin to prevent the formation of plasma. On the other side of the quartz disc, the outer aluminium electrode in the shape of a mesh was placed and connected to the power source ground. After the application of a high voltage AC signal, plasma was generated at the edges of aluminium mesh. The electrode unit was attached to the homemade power source, which operated at voltages 7 to 10 kV peak to peak at frequency of 40 kHz. 

Two plasma powers were used, termed low and high power, to generate contrasting plasma chemistries ranging from an ROS dominated mode (at low power) and an RNS dominated mode (at high power). A full description of the RONS produced under different operating power regimes can be found in our previous work [19,20,33].

During the treatments, the electrode unit was placed over the glass coverslips at fixed distance 5 mm between plasma ignition point and mycotoxin sample. The mycotoxins were treated for different time durations (30, 60, 120, 240, and 480 s). Following the exposures, the mycotoxins were washed from the surface of the glass coverslips with 0.5 mL of acetonitrile, and the solutions were transferred to vials for further LC-MS/MS analysis. For the comparison, mycotoxins were also irradiated with UVC lamp, considering equal treatment distance and exposure times. UVC irradiation was performed with standard UVC germicidal lamp (Philips, Amsterdam, The Netherlands), with highest intensity line of Hg at 253.65 nm (Appendix A). 

The experiments were performed in multiple repetitions to provide statistical significance, which was analysed with one-way ANOVA and nonlinear curve fit function (GraphPad Prism 8, GraphPad, San Diego, CA, USA).

### 4.2. LC-MS/MS Analysis of Treated Mycotoxins

Mycotoxins in solutions were quantified with LC-MS/MS (Waters, USA). All chromatographic separations were performed on a Zorbax Eclipse Plus C18 Rapid Resolution HD column, 2.1 × 100 mm, 1.8 µm (Agilent Technologies, Santa Clara, CA, USA). The chromatographic separation was performed by mixing of two mobile phase components (A and B) in isocratic mode (AFs) or gradient mode (trichothecenes, FBs, and ZEN). Component A was deionized water and component B was methanol, both containing 0.5% acetic acid and 2.5 mM ammonium acetate. Sample injection volume was 10 µL.

Isocratic elution of AFs was performed with a mixture of component A and B in the ratio 60:40. Sample was injected into column with mobile phase flow rate 0.3 mL/min and temperature 40 °C. Detection with MRM (multiple reaction monitoring) mode and ESI in positive ionization mode was performed in following conditions: source temperature 150 °C, desolvation temperature 200 °C, the capillary voltage 3.7 kV, the cone gas flow 20 L/h, and desolvation gas flow 800 L/h. For measuring the presence of AFB_1_, AFB_2_, AFG_1_, and AFG_2_, the ions at *m/z* 313, 315, 329, and 331 were monitored, respectively.

For the separation of other mycotoxins, time-programmed gradient was applied with initial composition 95% of A and 5% of B. The portion of component B was linearly increased to 40% within 4 min and further increased to 70% within the next 8 min. This latter composition was held for 4 min, and then component B was increased to 90% in 1.5 min. The proportion of component B was held at 90% for 2.5 min and then returned back to 5% in 1 min. The final composition was held for 4 min. The mobile phase flow rate was 0.3 mL/min, and the column temperature was 40 °C. MS/MS analysis was carried out in MRM mode. The mass spectrometer was operated in the electrospray ionization mode (ESI). During the run, ESI source was switching between positive (ESI+) and negative (ESI–) mode. Capillary voltage in ESI+ mode was 3.4 kV and in ESI–mode 3.0 kV. The desolvation temperature was 500 °C, the ion source temperature was 150 °C, and the collision cell voltage and cone voltages were optimized for each mycotoxin as shown in Table 3. 

The areas of the chromatogram peak of corresponding mycotoxin were measured, out of which the concentrations were calculated, using the standard curve obtained from the serial dilutions of standards. Considering the extraction of mycotoxins from food matrices, the concentrations were then converted from µg/mL (ppm) to µg/kg or mg/kg.

## Figures and Tables

**Figure 1 toxins-11-00219-f001:**
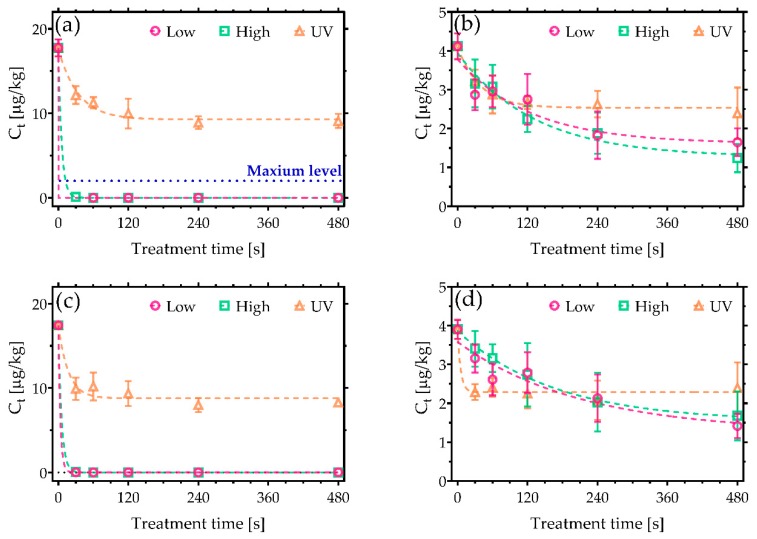
Cold Atmospheric pressure Plasma (CAP) decontamination of aflatoxins (AFs) under low and high discharge power conditions and a comparison against UVC treatment. Plots show the reduction in concentration of (**a**) AFB_1_, (**b**) AFB_2_, (**c**) AFG_1_, and (**d**) AFG_2_. The maximum permissible level of AFB_1_ allowed in cereals of 2 µg/kg is shown in (**a**) [21].

**Figure 2 toxins-11-00219-f002:**
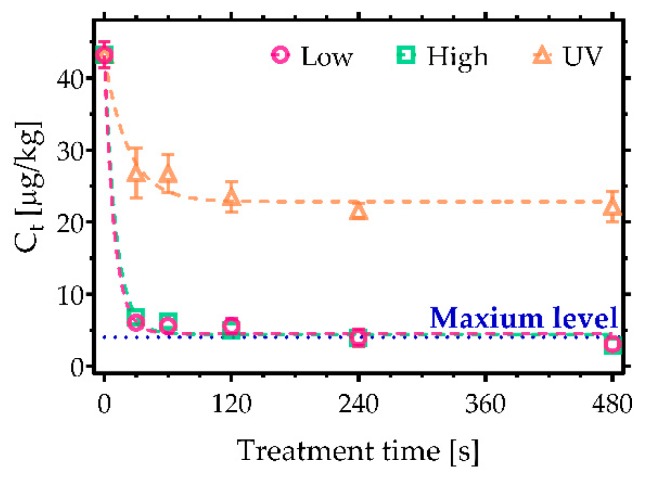
CAP decontamination of AFs considering the sum of their concentrations based on the air surface barrier discharge (SBD) plasma system operated at low and high discharge power. A comparison with UVC irradiation. Maximum level of AFs sum allowed in cereals is 4 µg/kg [21].

**Figure 3 toxins-11-00219-f003:**
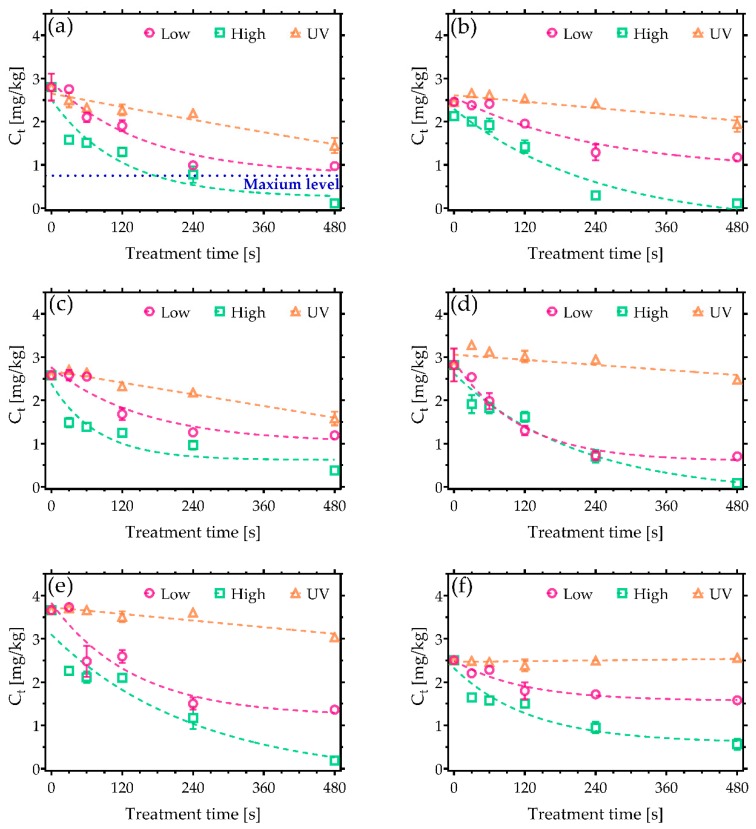
CAP decontamination of trichothecenes under low and high discharge power conditions and a comparison against UVC treatment. Plots show the reduction in concentration of (**a**) deoxynivalenol (DON), (**b**) 3-Acetyl DON (3-AcDON), (**c**) 15-Acetyl DON 15-AcDON, (**d**) diacetoxyscirpenol (DAS), (**e**) HT-2, and (**f**) T-2 as a function of CAP exposure time. The maximum permissible level of DON allowed in cereals of 0.75 mg/kg is shown on (**a**) [22].

**Figure 4 toxins-11-00219-f004:**
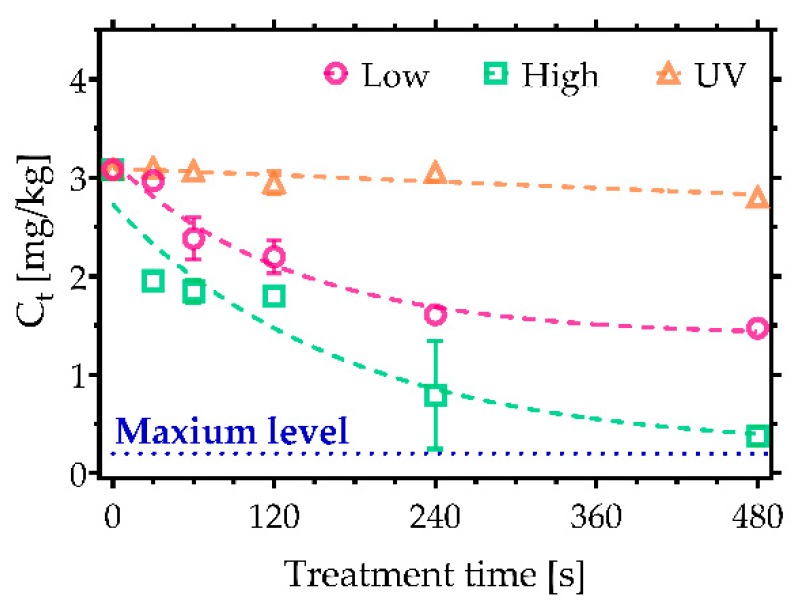
CAP decontamination of HT-2 and T-2 considering the sum of their concentrations based on the air surface barrier discharge (SBD) plasma system operated at low and high discharge power. A comparison with UVC irradiation. Maximum level of HT-2 and T-2 allowed in oats is 0.2 mg/kg [23].

**Figure 5 toxins-11-00219-f005:**
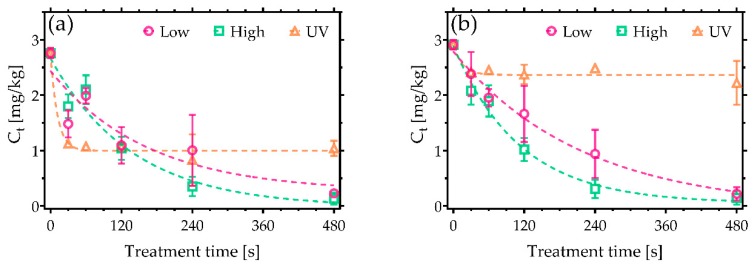
CAP decontamination of FBs under low and high discharge power conditions and a comparison against UVC treatment. Plots show the reduction in concentration of (**a**) FB_1_ and (**b**) FB_2_ as a function of exposure time.

**Figure 6 toxins-11-00219-f006:**
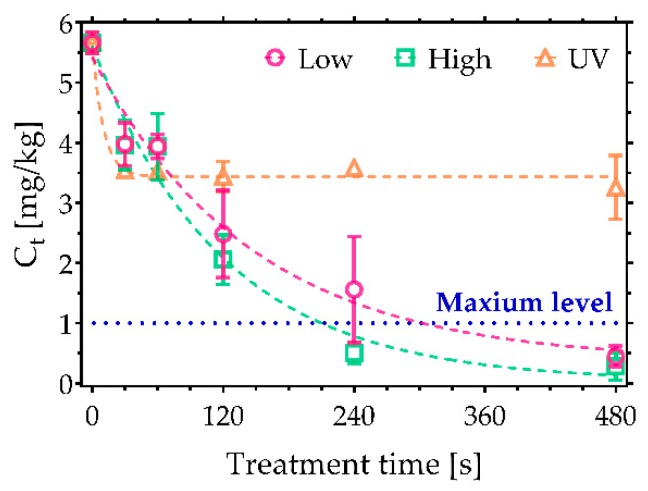
CAP decontamination of FBs considering the sum of their concentrations based on the air SBD plasma system operated at low and high discharge power. A comparison with UVC irradiation. Maximum level of FBs sum allowed in maize products is 1 mg/kg [22].

**Figure 7 toxins-11-00219-f007:**
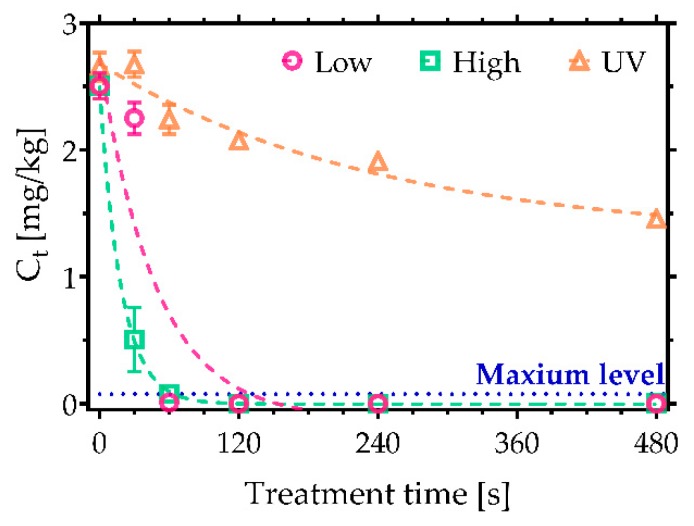
Decontamination of zearalenone (ZEN) based on the air SBD plasma system operated at low and high discharge power. A comparison with UVC irradiation. Maximum level of ZEN allowed in cereals is 0.1 mg/kg [22].

**Table 1 toxins-11-00219-t001:** Maximum levels of mycotoxins allowed in selected food products in the European Union (EU) [21,22,23].

Mycotoxin	Maximum Level Allowed [µg/kg]	Type of Food Product
AFB_1_/AFs	2/4	All cereals and all products derived from cereals, including processed cereal products, with the exception of maize; processed cereal-based foods and baby foods for infants and young children; and dietary foods for special medical purposes, intended specifically for infants
DON	750	Cereals intended for direct human consumption; cereal flour (including maize flour, maize meal, and maize grits); and bran as an end product marketed for direct human consumption and germ, with the exception of processed cereal-based foods and baby foods for infants and young children
HT-2 & T-2	200	Oats for direct human consumption
	2000	Oat milling products for feed (husks)
FBs	1000	Maize flour, maize meal, maize grits, maize germ, and refined maize oil
ZEN	200	Maize intended for direct human consumption, maize flour, maize meal, maize grits, maize germ, and refined maize oil

**Table 2 toxins-11-00219-t002:** The concentrations of the stock solutions of mycotoxins used in presented study.

Mycotoxin	Stock Solution Concentration	Acetonitrile/diH_2_O Ratio
*AFs*		
AFB_1_, AFG_1_	0.2 µg/mL	2:1
AFB_2_, AFG_2_	0.05 µg/mL	
*Trichothecenes*		
DON, 3-AcDON, 15-AcDON, DAS, HT-2, T2	27 µg/mL	2:1
*FBs*		
FB_1_, FB_2_	25 µg/mL	1:1
*ZEN*	27 µg/mL	2:1

**Table 3 toxins-11-00219-t003:** Tandem Mass Spectrometry (LC-MS/MS) detector parameters.

Mycotoxin	Ionization Mode	RT (min)	Precursor Ion (m/z)	Quantifier Ion (m/z)	Qualifier Ion (m/z)	Cone Voltage (V)	Collision Energy (V)
DON	ESI+	3.14	297.3	203.1	249.1	18/18	16/10
3-AcDON	ESI+	5.00	339.1	203.1	137.0	24/14	16/5
15-AcDON	ESI+	5.00	339.1	136.95	261.1	16/16	12/10
DAS	ESI+	7.06	384.3	307.2	247.2	20/20	14/10
HT-2	ESI+	8.91	442.4	215.1	263.2	12/12	16/14
T-2	ESI+	10.27	484.4	185.1	215.2	18/18	24/20
ZEN	ESI-	11.30	317.20	131.0	174.95	35/40	22/24
FB_1_	ESI+	10.20	722.4	334.2	352.2	30/30	40/35
FB_2_	ESI+	12.70	706.4	318.2	336.2	35/35	40/40
OTA	ESI+	11.10	404.2	221.0	239.0	24/24	36/24

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
