# Peer review of "Mycotoxin Decontamination Efficacy of Atmospheric Pressure Air Plasma"

_toxins, 2019, doi:10.3390/toxins11040219_

Reviewer 1 Report

The question is wheather the degradation of mycotoxins by CAP would be the same in contaminated food compared to results obtained. The work would have a much greater value if the authors extended their research to this issue.

Author Response

R: We would like to express our gratitude for taking time to review our manuscript and for giving us back a positive feedback. We also agree with the comment that presenting the results based on the CAP mycotoxin decontamination on real food cases would give a greater value to our research. At this point, we would like to clarify that the work shown in this manuscript is just a part of a large research scheme performed with an aim to wholly understand CAP decontamination process, a part of which is also investigation of mycotoxin decontamination efficacy of CAP performed on real food cases. However, we are in the process of submitting the results of latter in another publication. For this reason, we attached a small part of the results to show that our CAP system also efficiently removed aflatoxin B1 from food products such as corn (97 % decontamination). 

Reviewer 2 Report

Mycotoxin decontamination by atmospheric pressure air plasma (CAP) is an interesting and novel way of mitigating the risk of mycotoxin exposure.

The work is well presented and designed, but I would like to raise the following concerns:

The discussion is based on the reduction of mycotoxin levels from very high values to lower than the regulated levels after treatment. However, authors should take in consideration that unprocessed cereals may not be decontaminated with such high values (e.g., it may not be allowed to reduce DON levels from 2.7 mg/kg to below 0.75 mg/kg, since there is also a limit for unprocessed cereals!).

I would also ask authors to discuss, if CAP will only be effective in superficial contamination.

Finally, in a real food matrix, will authors predict any changes in other properties of the cereal?

Minor comments

-          Table 1 presents a few examples of the mycotoxin legislation in the EU. As it mention just a few examples, the information is misleading. For instance, the information reported for aflatoxins are for all cereals except maize and rice; the information on DON are for cereals after processing (but not all products, and unprocessed cereals have their own limits), … I suggest to have a more detailed description of the type of food product

-          I wonder in figure 2 is needed, since it is the sum of previous data!

-          Line 133: typing issue: 2-propane-1,2,3-tricarboxylic

-          Line 248-254: check how you present temperatures, should be 40 ºC, instead of 40º C.

-          - In M&M, it is not described how samples were treated before HPLC analysis. May be just one sentence, such as “vials were washed with mobile phase, … “

Author Response

Mycotoxin decontamination by atmospheric pressure air plasma (CAP) is an interesting and novel way of mitigating the risk of mycotoxin exposure.

The work is well presented and designed, but I would like to raise the following concerns:

The discussion is based on the reduction of mycotoxin levels from very high values to lower than the regulated levels after treatment. However, authors should take in consideration that unprocessed cereals may not be decontaminated with such high values (e.g., it may not be allowed to reduce DON levels from 2.7 mg/kg to below 0.75 mg/kg, since there is also a limit for unprocessed cereals!).

I would also ask authors to discuss, if CAP will only be effective in superficial contamination.

Finally, in a real food matrix, will authors predict any changes in other properties of the cereal?

R: We would like to thank the reviewer for taking a time to consider our manuscript and giving some helpful corrections and comments. We would also like to point out that the presented manuscript is just a tip of an ice-berg of performed research. The efficiency of the CAP removal of mycotoxins on real food products and negative effects of the CAP treatment on food products were one of our main concerns as well and a lot of our work was dedicated to the investigation of these areas. Nevertheless, the obtained results are the subject of another publication, which is in the process of submitting and is not yet published. Nevertheless, we attach a few results of the experiments which were conducted to determine the decontamination efficiency of CAP on AFB1 contaminated corn. As shown below, we reached around 97 % decontamination rate. Moreover, according to the FT-IR analysis, no modification of the surface of the corn kernel was recorded.

We are also aware of the fact that different maximum values of mycotoxins are prescribed for different types of food products. To avoid the confusion, we focused on the most common types.

Minor comments

1.          Table 1 presents a few examples of the mycotoxin legislation in the EU. As it mention just a few examples, the information is misleading. For instance, the information reported for aflatoxins are for all cereals except maize and rice; the information on DON are for cereals after processing (but not all products, and unprocessed cereals have their own limits), … I suggest to have a more detailed description of the type of food product

R: We would like to thank the reviewer for pointing this out. Table 1 has been modified.

2.          I wonder in figure 2 is needed, since it is the sum of previous data!

R: The legislation is prescribing the maximum values allowed separately for AFB1 and the sum of all AFs. In addition, the experiments were performed with the mixture of mycotoxins. For these reasons, we think that both of the variations have to be shown in separate graphs. 

3.         Line 133: typing issue: 2-propane-1,2,3-tricarboxylic

R: The typo has been corrected.

4.          Line 248-254: check how you present temperatures, should be 40 ºC, instead of 40º C.

R: The typo has been corrected.

5.        In M&M, it is not described how samples were treated before HPLC analysis. May be just one sentence, such as “vials were washed with mobile phase, … “

R: We thank the reviewer for exposing this. Additional information about HPLC analysis has been added.

Reviewer 3 Report

The manuscript is prepared accordingly to Toxins Journal with an interesting subject. The authors discussed new approaches of mycotoxin decontamination by using Cold Atmospheric Air Plasma.
The suggested minor changes should be considered before the acceptance of the manuscript:

1) In section: 4 Materials and methods

4.1. Preparation of mycotoxins and CAP treatments

- Explanation of used ratios of Acetonitrile/diH2O should be added

- Explanation how was the number of mycotoxins and efficiency of CAP treatment calculated

2) In Abstract:

In all cases, CAP was found to be considerably more efficient than UVC, with aflatoxin G1 and zearalenone being completely degraded, levels that could not be achieved using UVC irradiation

- should be highlighted that it is more efficient for „isolated” mycotoxins

3) In Conclusions:

In addition, the assessable system set-up enables the use of CAP, which is safe and low cost, with negligible effect on the environment

Safety, costs and effect on the environment were not part of this research, therefore I recommend to do not use it in conclusion. The Conclusion should be the summary of the Authors' research and statement (in italics) can be used in the Discussion part (including reference of this statement).

Overall remark:

The added value of the research could be to consider influence and effectiveness of CAP treatment of mycotoxins in food matrix to see whether CAP treatment can be applied to contaminated food with similar efficiency to this presented in research for the stock solutions of mycotoxins

Nevertheless, the overall paper is clearly presented, with promising results of the preliminary study for the CAP application.

Author Response

The manuscript is prepared accordingly to Toxins Journal with an interesting subject. The authors discussed new approaches of mycotoxin decontamination by using Cold Atmospheric Air Plasma.
The suggested minor changes should be considered before the acceptance of the manuscript:

We thank the reviewer for giving the work presented in our manuscript a positive review.

1) In section: 4 Materials and methods

4.1. Preparation of mycotoxins and CAP treatments

- Explanation of used ratios of Acetonitrile/diH2O should be added.

R: The explanation has been added.

- Explanation how was the number of mycotoxins and efficiency of CAP treatment calculated

R: The explanation was already in the manuscript (p.10, line 271-274).

2) In Abstract:

In all cases, CAP was found to be considerably more efficient than UVC, with aflatoxin G1 and zearalenone being completely degraded, levels that could not be achieved using UVC irradiation

- should be highlighted that it is more efficient for „isolated” mycotoxins

R: We demonstrated that CAP treatments reached a magnitude higher performance decontamination in comparison to UVC, no matter the type and mixture of mycotoxins. For this reason, we think that any additional highlight of the ‘isolated’ mycotoxins would not be necessary.

3) In Conclusions:

In addition, the assessable system set-up enables the use of CAP, which is safe and low cost, with negligible effect on the environment

Safety, costs and effect on the environment were not part of this research, therefore I recommend to do not use it in conclusion. The Conclusion should be the summary of the Authors' research and statement (in italics) can be used in the Discussion part (including reference of this statement).

R: We would like to thank the reviewer for expressing this. We removed the sentence concerning the safety of our CAP system.

Overall remark:

The added value of the research could be to consider influence and effectiveness of CAP treatment of mycotoxins in food matrix to see whether CAP treatment can be applied to contaminated food with similar efficiency to this presented in research for the stock solutions of mycotoxins

Nevertheless, the overall paper is clearly presented, with promising results of the preliminary study for the CAP application.

R: We agree with the comment that presenting the results based on the CAP mycotoxin decontamination on real food cases would give a greater value to our work. At this point, we would like to stress out that the results presented in this manuscript are just a part of an extensive research performed with an aim to wholly understand CAP decontamination process, a part of which is also investigation of mycotoxin decontamination efficacy of CAP performed on real cases such as corn kernels. However, we are in the process of submitting the results of latter in another publication. Nevertheless, in the figure below we show some of the results, which demonstrate that CAP treatments efficiently removed aflatoxin B1 from contaminated corn kernels (97% decontamination).

Round  2

Reviewer 1 Report

I accept the corrections.